# Neoadjuvant T-DM1/pertuzumab and paclitaxel/trastuzumab/pertuzumab for HER2+ breast cancer in the adaptively randomized I-SPY2 trial

Amy S. Clark [1][✉], Christina Yau[2], Denise M. Wolf[2], Emanuel F. Petricoin[3], Laura J. van 't Veer [2], Douglas Yee[4], Stacy L. Moulder[5], Anne M. Wallace[6], A. Jo Chien[2], Claudine Isaacs [7], Judy C. Boughey [8], Kathy S. Albain[9], Kathleen Kemmer[10], Barbara B. Haley[11], Hyo S. Han[12], Andres Forero-Torres[13], Anthony Elias[14], Julie E. Lang [15], Erin D. Ellis[16], Rachel Yung[17], Debu Tripathy [5], Rita Nanda [18], Julia D. Wulfkuhle[5], Lamorna Brown-Swigart [2], Rosa I. Gallagher [5], Teresa Helsten[6], Erin Roesch[6], Cheryl A. Ewing[2], Michael Alvarado[2], Erin P. Crane[7], Meredith Buxton[19], Julia L. Clennell[19], Melissa Paoloni[19], Smita M. Asare[2], Amy Wilson[2], Gillian L. Hirst [2], Ruby Singhrao[2], Katherine Steeg[2], Adam Asare[2], Jeffrey B. Matthews[2], Scott Berry[19], Ashish Sanil[19], Michelle Melisko[2], Jane Perlmutter[20], Hope S. Rugo [2], Richard B. Schwab [6], W. Fraser Symmans [5], Nola M. Hylton [2], Donald A. Berry[19], Laura J. Esserman [2] & Angela M. DeMichele[1]

HER2-targeted therapy dramatically improves outcomes in early breast cancer. Here we report the results of two HER2-targeted combinations in the neoadjuvant I-SPY2 phase 2 adaptive platform trial for early breast cancer at high risk of recurrence: ado-trastuzumab emtansine plus pertuzumab (T-DM1/P) and paclitaxel, trastuzumab and pertuzumab (THP). Eligible women have >2.5 cm clinical stage II/III HER2+ breast cancer, adaptively randomized to T-DM1/P, THP, or a common control arm of paclitaxel/trastuzumab (TH), followed by doxorubicin/cyclophosphamide, then surgery. Both T-DM1/P and THP arms 'graduate' in all subtypes: predicted pCR rates are 63%, 72% and 33% for T-DM1/P (n = 52), THP (n = 45) and TH (n = 31) respectively. Toxicity burden is similar between arms. Degree of HER2 pathway signaling and phosphorylation in pretreatment biopsy specimens are associated with response to both T-DM1/P and THP and can further identify highly responsive HER2+ tumors to HER2-directed therapy. This may help identify patients who can safely de-escalate cytotoxic chemotherapy without compromising excellent outcome.

[1] University of Pennsylvania, Philadelphia, PA, USA. [2] University of California San Francisco, San Francisco, CA, USA. [3] George Mason University, Fairfax, VA, USA. [4] University of Minnesota, Minneapolis, MN, USA. [5] MD Anderson Cancer Center, Houston, TX, USA. [6] University of California San Diego, San Diego, CA, USA. [7] Georgetown University, Washington, DC, USA. [8] Mayo Clinic, Rochester, MN, USA. [9] Loyola University, Chicago, IL, USA. [10] Oregon Health & Science University, Portland, OR, USA. [11] University of Texas Southwestern, Dallas, TX, USA. [12] Moffitt Cancer Center, Tampa, FL, USA. [13] University of Alabama Birmingham, Birmingham, AL, USA. [14] University of Colorado Denver, Aurora, CO, USA. [15] University of Southern California, Los Angeles, CA, USA. [16] Swedish Cancer Institute, Seattle, WA, USA. [17] University of Washington, Seattle, WA, USA. [18] University of Chicago, Chicago, IL, USA. [19] Berry Consultants, LLC, Houston, TX, USA. [20] Gemini Group, Ann Arbor, MI, USA. [✉]email: Amy.Clark@pennmedicine.upenn.edu

HER2-overexpressing breast cancer outcomes have improved dramatically in the last decade with the addition of HER2-directed therapy to chemotherapy for early breast cancer. In the neoadjuvant setting, pathologic complete response (pCR) rates are in the 40–50% range with taxane/trastuzumab combinations, and even higher with the addition of other HER2-targeted agents. pCR has also proven to be a strong surrogate of event-free survival for individual patients[1,2], particularly in hormone receptor-negative tumors. This provides multiple opportunities: to determine whether novel investigational agents can further improve pCR rates, assess comparative toxicity, and determine how increased pCR rates affect long-term survival outcomes. As such, the I-SPY2 trial is a neoadjuvant platform trial in which serial tumor samples are collected to identify biomarkers indicative of highly responsive or non-responsive tumors. These biomarkers are essential to optimizing therapy with strategies to de-escalate toxic treatment for highly responsive tumors and escalate therapy for those that are poorly responsive.

T-DM1 (ado-trastuzumab emtansine, Kadcyla) is an intravenous drug–antibody conjugate that links the HER2-targeted monoclonal antibody trastuzumab to emtansine, an active but systemically toxic chemotherapeutic[3,4]. T-DM1 is currently FDA-approved as a single agent for the treatment of patients with HER2[+], metastatic breast cancer who previously received trastuzumab and a taxane[5]. Pertuzumab (Perjeta) is a humanized monoclonal antibody that targets the extracellular dimerization domain of HER2, distinct from the binding site of trastuzumab. Pertuzumab is FDA-approved for use in combination with trastuzumab and docetaxel in metastatic breast cancer, and in combination with trastuzumab and chemotherapy as neoadjuvant or adjuvant therapy in non-metastatic disease.

This report describes the evaluation of two I-SPY2 trial arms: a neoadjuvant non-chemotherapy regimen, T-DM1 + pertuzumab (T-DM1/P), and a dual-HER2 targeting regimen, paclitaxel + trastuzumab + pertuzumab (THP), compared to paclitaxel + trastuzumab alone (TH) for HER2[+] breast cancer in the I-SPY2 trial. Moreover, we examined the potential for pre-specified multi-omic biomarkers reflecting degrees of HER2 pathway activation, estrogen receptor signaling, and proliferation to predict enhanced HER2-directed treatment response in patients already deemed HER2[+] by standard CLIA assays. These were examined in three signatures: the overall HER2[+] group, as well as the hormone receptor (HR)[+] HER2[+] and HR[−]/HER2[+] groups.

## Results

Fifty-two patients were enrolled in the T-DM1/P arm at 15 clinical sites from June 6, 2013 through August 17, 2015, when the arm graduated in all HER2[+] signatures (all patients, HR[+]/HER2[+], and HR[−]/HER2[+]). During the same time period, 45 patients were enrolled in the THP arm when the arm graduated in all HER2 signatures and was converted to the HER2[+] control arm. Thirty-one patients were randomized to the TH control arm across 17 sites from March 10, 2010 through to August 17, 2015 (Fig. 1). Baseline characteristics of the three arms were similar (Table 1), with the exception that the T-DM1/P arm had excess Mammaprint ultra-high (MP2) tumors (44%)

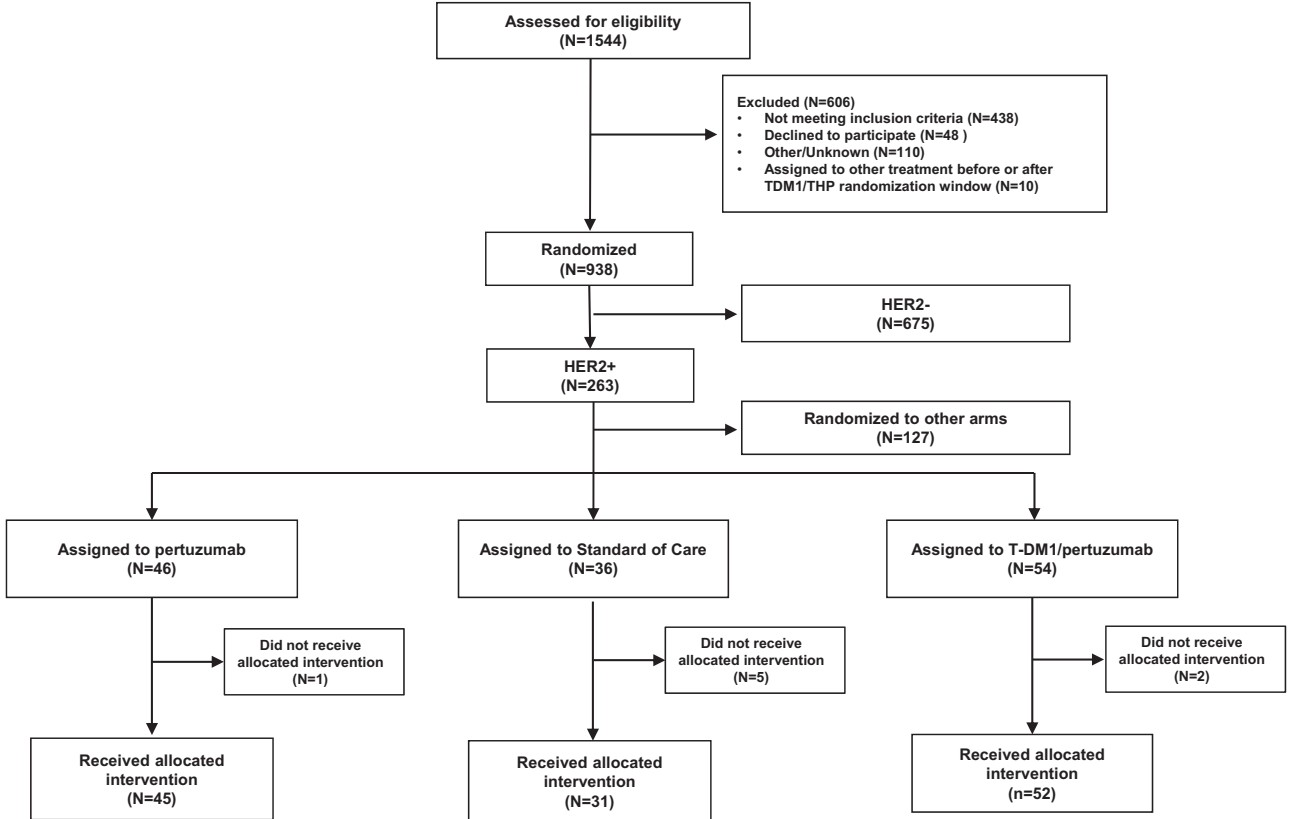

**Fig. 1 Consort diagram for the T-DM1/Pertuzumab, THP, and control populations.** Consort diagram shows the number of patients screening, randomized, and receiving allocated therapy from the start of I-SPY2 to the close of the T-DM1/Pertuzumab and THP arms. I-SPY2 is modified intent-to-treat, where patients receiving their allocated therapy are considered evaluable for analysis.

**Table 1 Demographics and baseline characteristics of participants.**

| Characteristic | T-DM1/P (n = 52) | THP (n = 45) | CONTROL (n = 31) |
|---|---|---|---|
| Median age, yr (range) | 48 (33-72) | 47 (29-70) | 50 (29-71) |
| Ethnicity, n (%) | | | |
| White | 42 (81%) | 37 (82%) | 25 (81%) |
| African American | 4 (8%) | 4 (9%) | 2 (6%) |
| Asian | 5 (10%) | 2 (4%) | 4 (13%) |
| Other/mixed | 1 (2%) | 2 (4%) | 0 (0%) |
| HR Status, n (%) | | | |
| Positive | 35 (67%) | 29 (64%) | 19 (61%) |
| Negative | 17 (33%) | 16 (36%) | 12 (39%) |
| Mammaprint, n (%) | | | |
| MP1 | 29 (56%) | 28 (62%) | 26 (84%) |
| MP2 | 23 (44%) | 17 (38%) | 5 (16%) |
| Median tumor size, cm (range) | 3.3 (1.5-12) | 3.4 (1.8-9) | 3.5 (1.3-11.7) |
| Baseline node status, n (%) | | | |
| Palpable | 18 (35%) | 17 (38%) | 10 (32%) |
| Non-palpable | 24 (46%) | 22 (49%) | 21 (68%) |
| N/A | 10 (19%) | 6 (13%) | 0 (0%) |
| HER2 qualifying test, n (%) | | | |
| IHC | 35 (67%) | 30 (67%) | 18 (58%) |
| FISH | 17 (33%) | 15 (33%) | 13 (42%) |
| HER2 IHC, n (%) | | | |
| IHC 3+ | 35 (67%) | 30 (67%) | 17 (55%) |
| IHC 2+ | 10 (19%) | 8 (18%) | 6 (19%) |
| IHC 1+ | 2 (4%) | 1 (2%) | 1 (3%) |
| Not reported | 5 (10%) | 6 (13%) | 7 (23%) |
| HER2 FISH, n (%) | | | |
| Positive | 29 (56%) | 25 (56%) | 15 (48%) |
| Equivocal | 0 (0%) | 0 (0%) | 1 (3%) |
| Negative | 1 (2%) | 0 (0%) | 1 (3%) |
| Not reported | 22 (42%) | 20 (44%) | 14 (45%) |
| HER2 TargetPrint, n (%) | | | |
| Positive | 39 (75%) | 32 (71%) | 18 (58%) |
| Negative | 13 (25%) | 13 (29%) | 10 (32%) |
| Not reported | 0 (0%) | 0 (0%) | 3 (10%) |

compared to control (16%; two-sided Fisher exact test $p = 0.01$). Source data is provided in Supplementary Data 1.

**Efficacy**. Figure 2 shows the pCR predicted probability curves (associated values are found in Supplementary Table 1). T-DM1/pertuzumab graduated in all three HER2[+] signatures. Amongst all HER2[+] patients in the TDM-1/P and TH arms, T-DM1/P had a higher estimated pCR rate (55%; 95% PI 41–69%) than TH (25%; 95% PI 11–38%), corresponding to a 99.9% probability that T-DM1/P was superior to TH and a 96% predictive probability of superiority in a 300-patient phase III trial. In the HR[+]/HER2[+] and HR[−]/HER2[+] signatures, the probability that pCR rates with T-DM1/P were superior to control were 99.7 and 98.8%, respectively.

THP also graduated in all three HER2[+] signatures, as shown in Fig. 2. Amongst all HER2[+] patients in the THP and TH arms, THP had a higher estimated pCR rate (56%; 95%PI 42–70%) than TH (25%; 95% PI 11–38%), corresponding to a 99.9% probability that THP was superior to TH and a 97% predictive probability of superiority in a 300-patients phase III trial. In the HR[+]/HER2[+] and HR[−]/HER2[+] signatures, the probability that pCR rates with THP were superior to control were 99.5 and 99.9%, respectively.

**Toxicity**. The toxicities of each arm are well described[6–8] and no new safety signals were seen in the I-SPY2 Trial arms (Table 2;

summary of all adverse events in Supplementary Data 2). Overall, adverse events are primarily grade 1/ 2 (see Supplementary Table 2) but one patient on the THP arm died due to respiratory failure during the THP portion of therapy and was unrelated to therapy. The majority of patients in each arm were able to receive AC after initial HER2-directed therapy (Table 2). Toxicities during AC differed between treatment arms: diarrhea and neutropenia were more frequent in the T-DM1/P arm, as was febrile neutropenia, while anemia was more frequent in the THP arm.

**Event-Free Survival (EFS)**. Follow-up was available for 47 (90%) T-DM1/P patients, 39 (86.7%) THP patients, and 31(100%) control patients, with median follow-up of 4.1 years. Over the follow-up period, a total of 7 events were observed in the T-DM1/P arm, 3 in the THP arm, and 7 in the common TH control arm. Notably, there were five CNS recurrences during the follow up period: three in the T-DM1/P arm, two in the TH arm, and none in the THP arm (Supplementary Fig. 1). Four of the five CNS recurrences observed were HR[−] (Supplementary Fig. 1).

Three-year EFS was 88% (95% CI: 79–99%) for T-DM1/P, 92% (95% CI: 84–100%) for THP and 87% (95% CI: 75–100%) for TH (Supplementary Fig. 1). In the two arms with brain metastases, EFS estimates were similar for patients who achieved pCR and those who did not: 88% regardless of pCR status for TDM-1/P (95% CI: 77–100% and 74–100% for pCR and non-pCR, respectively), and 88% (67–100%) vs 86% (73–100%) for pCR vs. non-pCR, respectively, in the TH control arm. For THP, EFS was 96% (89–100%) vs 85% (68–100%) for pCR vs non-pCR.

**Biomarker assessment**. We assessed 10 biomarkers in the HER2, ER/PR, and proliferation pathways as predictors of response to TDM1/P and THP, hypothesizing that highly HER2-activated, non-luminal A or highly proliferative tumors may be more sensitive to anti-HER2 therapies than those that are less HER2-activated, more luminal or quiescent. HER2 and HER2 signaling was evaluated at 5 levels of resolution: IHC,quantitative protein, and phosphoprotein measurements by RPPA (total ERBB2; phosphorylated [p] pERBB2, phosphorylated [p] pEGFR), and mRNA (HER2 amplicon module Module7_ERBB2). These HER2 biomarkers were highly correlated (Supplementary Fig. 2A) and all five HER2 pathway biomarkers were significantly associated with pCR in the TDM1/P and THP arms (Fig. 3A–G, LR $p < 0.05$ as detailed in Table 4). Nearly all associations retained significance in a model adjusting for HR status, and in HR[+]/HER2[+] subsets (Table 3, Supplementary Fig. 2C–H). HER family (HER2 and EGFR) activation/phosphorylation signatures clustered distinctly from total HER2 as measured by mRNA and RPPA (Supplementary Fig. 2A). Plotting pEGFR and pERBB2 expression together revealed a separation into two groups of patients: one with excellent response and one without (Fig. 3G). Since HER2 and EGFR are well-known heterodimerization partners, the fact that both are found to be highly co-activated in the pre-treatment tumor samples of responding patient population provide further evidence of functional HER2-driven pathway activation and signaling coherence.

ER/PR signaling, represented by the average expression of ESR1 and PGR, was also evaluated. Higher ER signaling levels were associated with non-response to TDM1/P and THP in the population as a whole (Fig. 4A, B, Table 3) and in the HR[+]/HER2[+] subsets within both arms (Table 3, Supplementary Figure S2J–K). Consistent with the above, tumors classified Luminal A by PAM50 had a lower pCR rate relative to those of other classes (Fig. 4C, Supplementary Fig. 2).

Finally, we quantitatively assessed proliferation markers at the total protein (RPPA: Ki67), phospho-protein (pAURKA) and

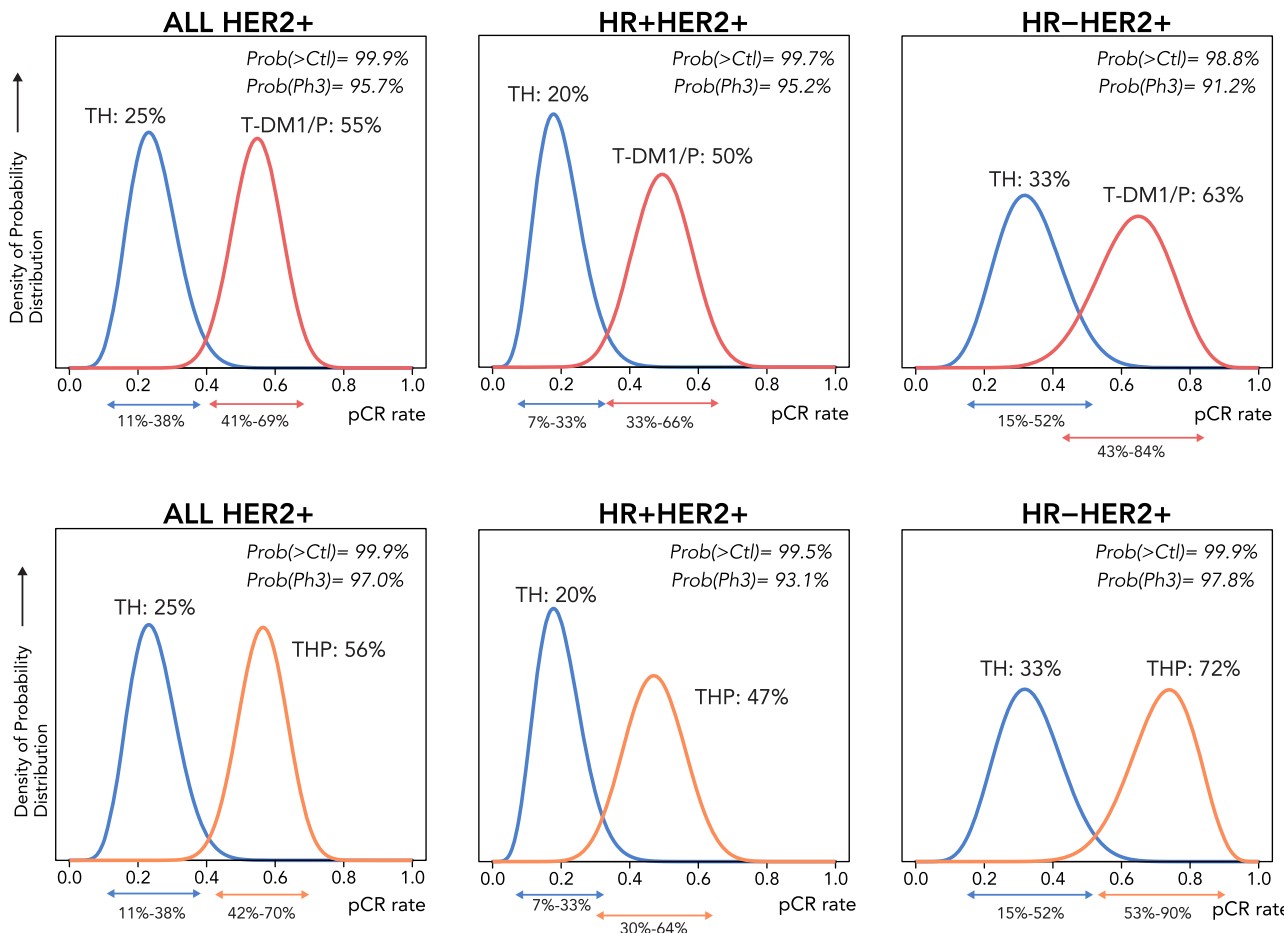

**Fig. 2 Primary efficacy analysis.** pCR probablility distribution curves of TDM1/P (red) vs TH (blue) and THP (orange) vs TH (blue). Arrows below x-axis indicate 95% probability interval derived from the I-SPY2 Bayesian time and covariate-adjusted logistic model described in the methods.

**Table 2 Adverse events Grade 3 and above, experienced by ≥5% of participants following TDM1, THP, TH, or AC.**

| | T-DM1+P arm | | THP arm | | TH control arm | |
|---|---|---|---|---|---|---|
| | T-DM1/P (n = 52) | AC (n = 49) | THP (n = 45) | AC (n = 40) | Paclitaxel (n = 31) | AC (n = 28) |
| Adverse event, n (%) | | | | | | |
| Anemia | 2 (3.9%) | 2 (4.1%) | 1 (2.2%) | 4 (10.0%) | 1 (3.2%) | 0 (0.0%) |
| Diarrhea | 0 (0.0%) | 6 (12.2%) | 0 (0.0%) | 0 (0.0%) | 1 (3.2%) | 0 (0.0%) |
| Febrile neutropenia | 0 (0.0%) | 8 (16.3%) | 0 (0.0%) | 5 (12.5%) | 0 (0.0%) | 3 (10.7%) |
| Hypertension | 2 (3.8%) | 0 (0.0%) | 2 (4.4%) | 0 (0.0%) | 3 (9.7%) | 3 (10.7%) |
| Neutrophil count decreased | 0 (0.0%) | 6 (12.2%) | 2 (4.4%) | 1 (2.5%) | 1 (3.2%) | 2 (7.1%) |
| Vascular access complication | 0 (0.0%) | 0 (0.0%) | 0 (0.0%) | 0 (0.0%) | 0 (0.0%) | 2 (7.1%) |
| White blood cell count decreased | 0 (0.0%) | 2 (4.1%) | 0 (0.0%) | 0 (0.0%) | 0 (0.0%) | 2 (7.1%) |
| Dose Reductions, n (%) | 1 (1.9%) | 3 (5.8%) | 3 (6.7%) | 5 (20%) | 0 (0.0%) | 0 (0.0%) |
| Early Discontinuation, n (%) | | | | | | |
| All | 3 (5.8%) | 5 (10.2%) | 5 (11%) | 4 (10%) | 3 (9.7%) | 3 (10.7%) |
| Toxicity | 1 (1.9%) | 3 (6.1%) | 3 (6.7%) | 2 (5.0%) | 0 (0.0%) | 0 (0.0%) |
| Progression | 1 (1.9%) | 0 (0.0%) | 0 (0.0%) | 0 (0.0%) | 0 (0.0%) | 0 (0.0%) |
| Other | 1 (1.9%) | 2 (4.1%) | 2 (4.4% | 2 (5.0%) | 3 (9.7%) | 3 (10.7%) |
| Median time to surgery*, days (range) | 170 (148–239) | | 176 (112–219) | | 171 (119–239) | |

mRNA (proliferation signature Module11_Proliferation) levels as predictors of response. In the population as a whole, the mRNA proliferation signature was associated with response in both T-DM1/P and THP, whereas pAURKA was associated with response in only THP. However, all three proliferation biomarkers significantly associate with response to TDM1/P but not THP in the HR+/HER2+ subset (Fig. 4D–F; Table 3).

Of the 10 markers evaluated, the proliferation markers showed the largest predictive performance differences between arms, particularly in the HR+/HER2+ subtype. Other differences include more dramatic HER2-pathway biomarker associations with pCR in the TDM1/P arm relative to THP. Examples include HER2 3+ IHC: OR = 7.1 [1.4–50] vs. 5.2 [0.9–40]; PAM50 HER2: OR = 21 [4–219] vs 1.6 [0.4–6.4]; pERBB2 amplicon:

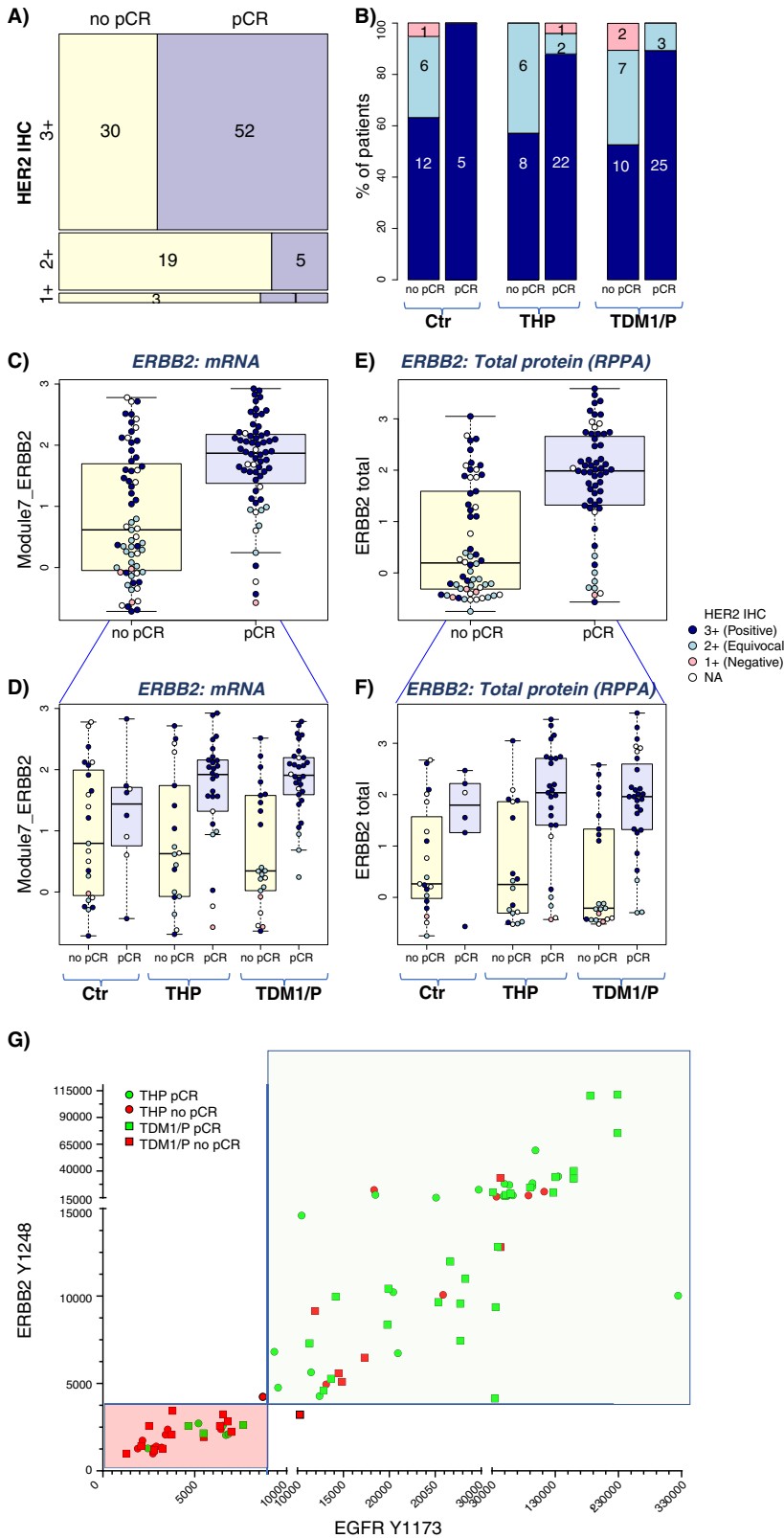

OR/unit increase = 5.5 [2.4–16] vs 2.3 [1.2–6.4]; pERBB2: OR/unit increase = 16.3 [2.6–245] vs. 5.5 [1.5–31]) and is further illustrated by scatter plots of pERBB2 vs. pEGFR by arm (Supplementary Fig. 3). Within the common control arm, of the 10 biomarkers tested only HER2 IHC 3+ significantly associates with response (LR $p = 0.046$), potentially due in part to small sample size (Table 3).

## Discussion

In HER2$^+$ early breast cancer, neoadjuvant treatment with T-DM1/P or THP demonstrated significant improvement in predicted pCR rates over treatment with TH, when all regimens were followed by AC and surgery. Similar improvement with T-DM1/P and THP over TH was observed in both HR$^+$ and HR$^-$ subsets, despite overall higher response rates for HR$^-$ disease.

**Fig. 3 Associations between HER family mRNA-, protein- and phosphoprotein-based pathway activation and pCR. A** Mosaic plot showing the proportion of patients who had pCR (purple) or did not have pCR (yellow) as a function of HER2 IHC level (1+, 2+, 3+; left to right), in the TDM1/P (top), THP (center), and control (bottom) rows. Barplots in **B** show these data by arm. Pink: IHC 1+; Light Blue: IHC 2+; Blue: IHC 3+. Panels **C**, **D** show response-association boxplots of a HER2 mRNA signature overall (**C**; $n = 127$) and by arm (**D**; $n = 52$ (TDM1/P), 44 (THP), and 31 (Ctr)) Yellow box is non-pCR, light blue box is pCR. Color of data points correspond to IHC level as in panel **B**. Panels **E**, **F** show response-association boxplots of HER2 total protein overall (**E**; $n = 117$) and by arm (**F**; $n = 49$ (TDM1/P), 43 (THP), and 25 (Ctr)). Yellow box is non-pCR, light blue box is pCR. Color of data points correspond to IHC level as in panel **B**. **G** Two-way scatter plot of phosphorylated HER2 (Y1248) (y-axis) and phosphorylated EGFR (Y1173) (x-axis) relative intensity (RI) values generated from the LCM-RPPA data in the pretreatment biopsy samples are shown along with pCR YES (green) and pCR NO (red) designations. For box plots, center line is group median; upper and lower limits of the box correspond to the 1st and 3rd quartile with whiskers extending to 1.5 times the interquartile range from top/bottom of the box.

Clinically, these results have been corroborated by several other studies and the overall findings with regard to pCR rates and toxicity are similar to what is seen in the I-SPY2 Trial. In fact, THP followed by AC became a care standard for neoadjuvant therapy in HER2+ breast cancer during its assessment in I-SPY2, with the FDA approval of pertuzumab based on results from multiple neoadjuvant trials[7,9,10]. T-DM-1/P alone, however, has not been adopted as a care standard largely because the KRISTINE trial showed better pCR rates with neoadjuvant taxane/platinum with HP (55.7%) over TDM-1/P (44.4%)[6]. Our pCR estimates in the TDM1/P arm are higher, likely because patients also received AC.

Critically, however, the I-SPY2 trial adds important and novel information to the field in the assessment of biomarkers that differentiate responsiveness to these Her2-directed therapies. Despite all tumors in these arms being classified as HER2+ by standard IHC and/or FISH, pre-treatment HER2 levels (whether measured by IHC, quantitative protein or gene expression), and activation of HER2 signaling (as measured by phosphorylation and co-activation of HER2 and EGFR, or PAM50 HER2 subtype), were all strong predictors of pathologic response to both TDM-1 and THP. Conversely, highly hormone-driven tumor biology as defined by luminal A subtype and low proliferation was inversely correlated with response in both experimental arms. Low levels of expression by protein and mRNA predict a much lower response to HER2 targeted therapy. These findings are critically important to the design of the next generation of HER2+ trials in which these biomarkers can be utilized to assess which patients can potentially de-escalate therapy (by dropping toxic anthracycline or platinum agents) from those who could benefit from treatment escalation with new and more intensive treatment approaches as illustrated in Fig. 5.

I-SPY2 is a biomarker-rich trial where we uniquely apply rigorous pre-specified statistical assessment to pre-defined mechanism-of-action biomarkers that span the DNA–RNA–protein–phosphoprotein biochemical landscape. The design helps us to learn why some patients respond and other do not. The phosphoprotein data generated here provides evidence for a phospho-HER2-EGFR cut point that, once validated, could guide escalation (those predicted to have a pCR) and de-escalation (those predicted to not have a pCR) of therapy (Fig. 3G). Our observation that HR status and estrogen signaling are associated with lower responsiveness to TDM1/P and THP is consistent with previous reports that luminal-type tumors are less responsive to HER2-targeted agents[11]. As both HER2 and ER biomarkers predict response to both arms, it is unlikely that either pathway biomarker alone would be sufficient to specifically predict response to TDM1/P vs. THP. However, we did observe a more dramatic signal for the HER2 pathway (particularly the pERBB2 and pEGFR) in the TDM1/P arm.

Recently, there has been a resurgence of interest to redefine breast cancer subclasses that respond to HER2-targeted therapies beyond the current repertiore of FDA-approved IHC and FISH

testing parameters[12]. This interest is underpinned by the acknowledged overall lack of clinical sensitivity and specificity of IHC and FISH to accurately idenfity those patients destined for response to HER2-targeted therapy including those both currently defined as HER2+ and HER2–[13]. Our qualifying gene/protein/phosphoprotein biomarkers were found to significantly associate with response in both T-DM1/P and THP arms, and differentiated HER2 IHC 3+ patients who did not achieve pCR as well as HER2 IHC 1/2+ patients who did not achieve pCR (Fig. 3). In other words, these biomarkers may help with escalation and descalation of HER2-directed therapy. For example, patients whose tumors are HER2 low/indeterminate (0-2+ by IHC) but have high phospho EGFR and HER2 levels) could be enriched for response to FDA-approved/experimental HER2 targeting agents. However, patients whose tumors are HER2 high (3+) but have low phospho EGFR and HER2 levels should potentially receive other experimental therapies combined with HER2-directed agents. Such combinations can be ethically explored in such patients because they are likely non-responsive to current FDA-approved HER2 targeted regimes. While there is a wealth of data that supports using TDM1 as rescue therapy in patients who have residual disease following neoadjuvant therapy[14], our data support further investigation of neoadjuvant TDM1/P specifically in HR+/HER2+ selected based on proliferation signatures. This approach could be particularly appealing for those in whom myelosuppression is a particular concern.

The major difference in predictive signal between arms were observed in the proliferation biomarkers in the HR+HER2+ subset, where these biomarkers associated with response in TDM1/P but not THP arm. If confirmed in independent studies, a combination of proliferation and pERBB2/pEGFR may help distinguish likely response to TDM1/P vs. THP in context of HR status. Taken together, these findings suggest that there is a great degree of heterogeneity amongst HER2-overexpressing breast cancers in the I-SPY population of patients with tumors >2.5 cm, which could potentially be exploited for patient selection strategies.

There are several limitations to the current analysis that impact interpretation, most notably the impact of small sample sizes that limit evaluation of both long-term outcomes and biomarkers. The benefit of the adaptive randomization in I-SPY2 is its efficiency in assigning the agents to the subsets of patients that benefit most, and drugs leave the trial when graduation is reached—that is, the minimum number of patients necessary can be exposed to investigational agents in order to determine signal in a statistically robust manner. With this greater degree of efficiency in evaluating pCR rates comes sample size/power limitations with respect to assessing EFS benefits or predictive biomarkers. Importantly, approximately 9% of the patients in KRISTINE who did not have a pCR on the T-DM1 arm received adjuvant chemotherapy, and there were 15 locoregional events prior to surgery in the T-DM1/P arm, suggesting the presence of a subgroup of patients with disease that was primarily refractory to HER2 antibody–drug

**Table 3 Biomarker pCR association results summary.**

| Biomarker (sig) | Type | All HER2+ subtypes — TDM1/P (Exp: n=52; RPPA: n=49; IHC: n=47) M1: pCR - sig OR/unit increase | LR p | Adjust for HR M2: pCR - sig – HR OR/unit increase | LR p | THP (Exp: n=44; RPPA: n=43; IHC: n=39) M1: pCR - sig OR/unit increase | LR p | Adjust for HR M2: pCR - sig +HR OR/unit increase | LR p | Ctr (TH) (Exp: n=31; RPPA: n=25; IHC: n=24) M1: pCR - sig OR/unit increase | LR p | Adjust for HR M2: pCR - sig OR/unit increase | LR p | HR+HER2+ subset — TDM1/P (Exp: n=35; RPPA: n=34; IHC: n=32) M1: pCR - sig OR/unit increase | LR p | THP (Exp: n=29; RPPA: n=29; IHC: n=25) OR/unit increase | LR p |
|---|---|---|---|---|---|---|---|---|---|---|---|---|---|---|---|---|---|
| ER_PGR_avg | Continuous - mRNA | 0.413 [0.19–0.809] | **0.00893** | 0.228 [0.05–0.734] | **0.0109** | 0.243 [0.0921–0.531] | **0.000165** | 0.14 [0.0233–0.494] | **0.000939** | 0.487 [0.198–1.01] | 0.0538 | 0.542 [0.156–1.55] | 0.262 | 0.0985 [0.00976–0.473] | **0.00146** | 0.164 [0.0279–0.561] | **0.00214** |
| Mod7_ERBB2 | Continuous - mRNA | 5.47 [2.43–15.6] | **4.84E-06** | 5.19 [2.36–14.5] | **5.71E-06** | 2.34 [1.24–4.88] | **0.00763** | 2.33 [1.2–4.96] | **0.0111** | 1.32 [0.593–3.1] | 0.5 | 113 [0.485–2.7] | 0.774 | 5.03 [1.99–16.9] | **0.000244** | 3.97 [1.61–13.2] | **0.00153** |
| Module11_Prolif_score | Continuous - mRNA | 2.18 [1.19–4.36] | **0.0111** | 2.11 [1.08–4.47] | **0.0293** | 2.55 [1.24–6.39] | **0.00861** | 2.4 [1.1–6.47] | **0.0254** | 1.77 [0.593–3.1] | 0.201 | 1.43 [0.485–2.7] | 0.471 | 3.28 [1.45–9.15] | **0.00311** | 1.63 [0.693–4.41] | 0.268 |
| ERBB2.total | Continuous - RPPA | 3.99 [1.96–9.51] | **4.56E-05** | 3.92 [1.92–9.31] | **6.02E-05** | 2.84 [1.46–6.18] | **0.00149** | 2.64 [1.33–5.88] | **0.00484** | 2.36 [0.794–8.81] | 0.125 | 1.98 [0.603–7.73] | 0.261 | 4.21 [1.77–12.8] | **0.000552** | 2.8 [1.23–7.58] | **0.0123** |
| ERBB2 Y1248 | Continuous - RPPA | 16.3 [2.56–245] | **0.000167** | 15.9 [2.64–217] | **0.00016** | 5.52 [1.45–31] | **0.00903** | 4.73 [1.23–25.4] | **0.0213** | 1.63 [0.597–4.61] | 0.32 | 1.08 [0.314–3.7] | 0.896 | 9.13 [1.78–113] | **0.00257** | 3.57 [0.837–20.7] | 0.0873 |
| EGFR Y1173 | Continuous - RPPA | 109 [5.96–5570] | **5.69E-06** | 123 [6.66–6230] | **3.67E-06** | 2.6 [1.01–10.6] | **0.0465** | 2.53 [0.949–9.73] | 0.0672 | 1.54 [0.492–4.6] | 0.43 | 0.979 [0.256–3.5] | 0.973 | 52.8 [3.49–2910] | **1.00E-04** | 2.12 [0.707–8.23] | 0.185 |
| Ki67 total | Continuous - RPPA | 1.24 [0.707–2.39] | 0.458 | 1.2 [0.677–2.31] | 0.535 | 1.83 [0.94–4.82] | 0.0798 | 1.56 [0.758–4.19] | 0.25 | 4.3 [0.809–32.5] | 0.0872 | 2.93 [0.437–26.9] | 0.277 | 7.15 [1.62–73.7] | **0.00286** | 0.961 [0.311–2.88] | 0.941 |
| Aurora A T288/B T232/C T198 | Continuous - RPPA | 1.3 [0.728–2.53] | 0.384 | 1.24 [0.683–2.46] | 0.485 | 4.75 [1.56–20.5] | **0.00364** | 4.22 [1.27–18.9] | **0.0171** | 1.06 [0.432–2.2] | 0.885 | 0.998 [0.31–2.16] | 0.996 | 4.01 [1.43–15.3] | **0.006** | 1.94 [0.421–10.3] | 0.396 |
| HER2 IHC (3+ vs. 2+/1+) | Categorical - IHC | 7.14 [1.4–50] | **0.0046** | 6.82 [1.6–36] | **0.0081** | 5.22 [0.86–40] | **0.031** | 3.98 [0.78–24] | 0.097 | n.a. | **0.046** | n.a. | 0.08 | 7.95 [1.17–96] | **0.0096** | 5.11 [0.64–68] | 0.060 |
| PAM50 subtype (HER2 vs others) | Categorical - mRNA | 20.6 [3.8–219] | **6.25E-06** | 41 [6.2–884] | **1.0E-05** | 1.55 [0.4–6.4] | 0.46 | 0.77 [0.16–3.3] | 0.73 | 4.5 [0.58–38] | 0.08 | 3.2 [0.45–24] | 0.24 | n.a. | **9.3E-04** | 1.09 [0.16–7.7] | 0.91 |

Bold indicates LR p < 0.05. *Because of the low number of patients with biomarker data who achieved pCR in the HR+HER2+ subset of the control arm (n < 4 pCR of 19 (Exp), 15 (RPPA), and 16 (IHC)), we do not calculate p-values.

conjugate-based therapy alone due to the inability to internalize the antibody–toxin, as suggested by Weyergang et al. (submitted). These findings reinforce our I-SPY2 results suggesting that HER2- and HR-related biology of the primary tumor play a large role in responsiveness to T-DM1, and confirming that adjuvant therapy must be considered when examining EFS, with more aggressive adjuvant therapy in non-pCR patients potentially improving outcomes preferentially for this group of patients compared to those with pCR, creating an equalizing effect.

The similar outcomes of patients with and without a pCR that we observed is likely to be related to adjuvant therapy received. Because the primary endpoint of I-SPY2 is pCR, the type or extent of adjuvant therapy is not mandated, and was at the discretion of the treating physican. Adjuvant therapies differed significantly between the three arms and between pCR and non-pCR groups (Supplementary Table 3). For example, seven patients on the non-taxane containing T-DM1/P arm received additional chemotherapy with adjuvant taxane plus trastuzumab, and all of these were in the non-pCR group. No patients in the THP or TH arms received adjuvant chemotherapy.

The CNS is a known sanctuary site for HER2+ breast cancer, even in the setting of pCR, a finding that is further corroborated in our study: the arms where CNS recurrences were found, difference in 3-year EFS between those with pCR vs. non-pCR was not seen. It is difficult to say whether the lack of brain metastases in the THP arm was due to chance, but previous reports indicate that the addition of pertuzumab to trastuzumab may delay onset of CNS disease[15]. Tucatinib, a recently FDA-approved drug has shown efficacy in metastatic HER2+ breast cancer patients with brain metastases[16], will be studied in the post-neoadjuvant setting combined with T-DM1 in the COMPASS-RD (NCT04457596) trial. Combining tucatinib with a less toxic regimen such as T-DM1 and avoiding the toxicity of AC could prove to be a highly effective intervention for women with tumors that highly express HER2 or fall into the HER2 molecular subtype. Development of drugs that penetrate the CNS and prevent CNS recurrence in HER2+ breast cancer is urgently needed to improve outcomes in these high-risk patients.

In summary, both T-DM1/P and THP significantly increased estimated pCR rates over TH in HER2+ patients, within both HR+ and HR− groups. Given similar high pCR rates and toxicity burden in the populations overall, it is important to develop tools that can be applied to individual patients to aid in the selection of the most active therapy in the clinic. Pretherapy gene expression and protein signaling further identifies highly sensitive tumors beyond those that are HER2+ by standard ASCO/CAP guidelines. The goal of I-SPY2 is to identify active agents, and generate useful guidance regarding the biological profiles of patients in whom further studies of an active agent or strategy should be employed. A high percentage of patients can achieve pCR with TDM1/P or THP. Without AC, these could be less toxic combinations that could be effective for HER2 molecular classification and proliferative tumors. The ability to eliminate AC altogether is under investigation in the COMPASS-pCR trial, where patients are treated to receive THP alone neoadjuvantly. The data reported here are useful in planning further definitive studies that will enable optimization of HER2-directed therapy for future phase III trials and shed important light on the heterogeneity of HER2+ breast cancer and its impact on response to targeted therapy.

## Methods

**Study design.** The I-SPY2 Trial (NCT01042379) is an ongoing phase 2 multi-center, open-label, adaptively randomized platform trial of neoadjuvant therapy (NAT) for early breast cancer that evaluates multiple investigational agents in parallel against a common control to evaluate their response within specific breast cancer subtypes (Supplementary Fig. 4). Biomarker assessments at screening are

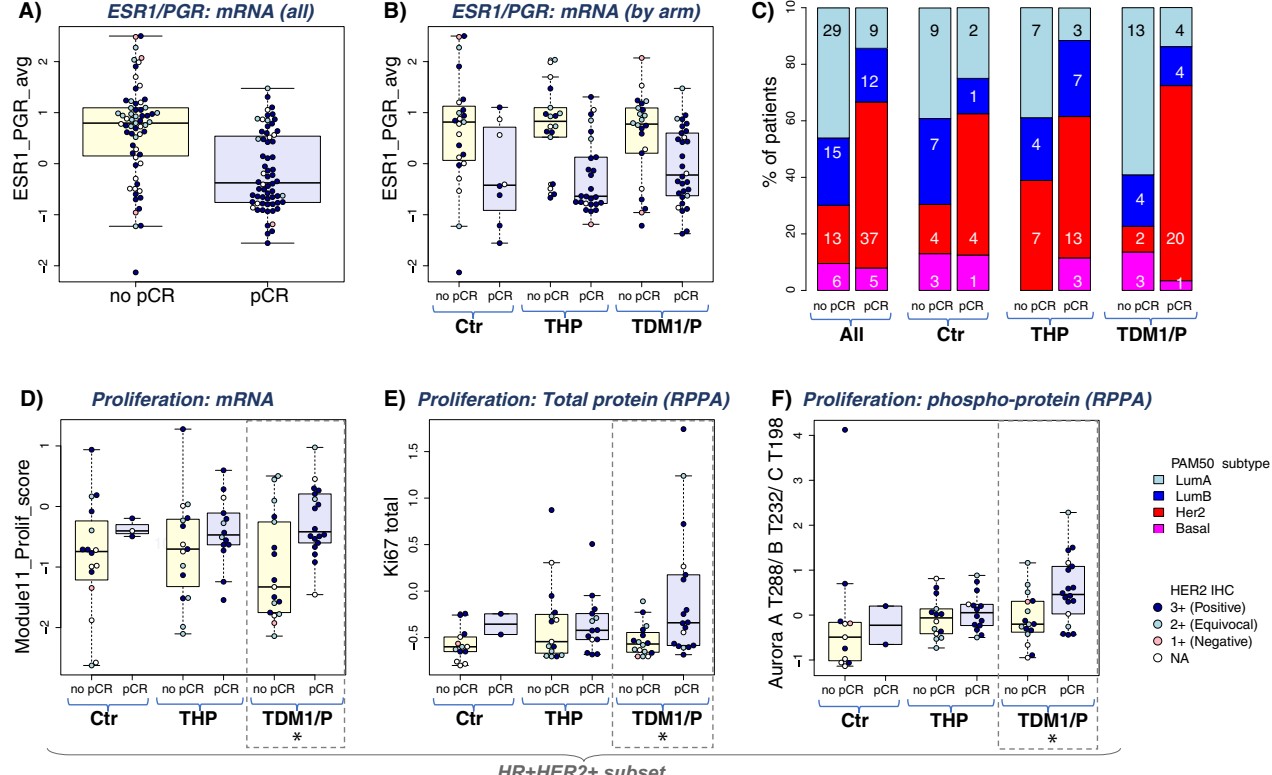

**Fig. 4 Associations between HR/luminal and proliferation biomarkers and pCR. A, B** shows HR expression (ESR1 and PGR averaged) response-association boxplots in all patients (**A**; $n = 127$) and by arm (**B**; $n = 52$ (TDM1/P), 44 (THP), and 31 (Ctr)). For all box plots, yellow box indicates non-pCR, light blue indicates pCR. **C** Bar plot showing the prevalence of PAM50 subtypes (LumA: light blue; LumB: dark blue; HER2:red; Basal: magenta) in patients achieving pCR compared to non-responders in the population as a whole (left pair of bars, n=126) and by arm (right pairs, n = 51 (TDM1/P), 44 (THP), and 31 (Ctr) n = 51 (TDM1/P), 44 (THP), and 31 (Ctr)). **D–F** show response-association boxplots of proliferation biomarkers on the mRNA (**D**; $n = 83$), protein (**E**; $n = 78$) and phospho-protein (**F**; $n = 78$) levels in the HR+/HER2+ subset by arm. *Association is significant (LR $p < 0.05$) only for TDM1/P, broken rectangle. For box plots, center line is group median; upper and lower limits of the box correspond to the 1st and 3rd quartile with whiskers extending to 1.5 times the interquartile range from top/bottom of the box.

used to assess eligibility and classify patients into one of eight subtypes based on hormone receptor (HR), HER2-receptor, and Mammaprint status[17]. The adaptive randomization engine preferentially assigns patients to agents based on continually updated Bayesian probabilities of pCR rates within predefined biomarker signatures; 20% of patients are randomized to the control arm.

The primary endpoint is pCR, defined as complete resolution of invasive cancer in both breast and lymph nodes (ypT0/is and ypN0). An investigational agent "graduates" from I-SPY2 if/when it achieves a ≥85% Bayesian predictive probability of demonstrating superiority to control in a hypothetical 1:1 randomized 300-patient phase 3 neoadjuvant trial in at least one of 10 clinical signatures[18,19]. Once the graduation threshold or a predetermined maximum enrollment is reached, accrual to the arm stops and predictive probabilities are updated after all patients complete surgery; note that the number of patients per arm is not fixed. Patients are followed for long-term outcome, including recurrence and death. Additional details on the study design have been published previously[20–22].

**Eligibility.** Adult women with stage II or III breast cancer with primary tumors ≥ 2.5 cm clinically or ≥ 2.0 cm by imaging who have not received prior treatment for their breast cancer are eligible for I-SPY2. All patients provide written informed consent at screening and again after randomization. Only patients with HER2+ disease as defined by ASCO/CAP guidelines were eligible for arms in the current report. Although TargetPrint HER2 expression was obtained for all patients on the three arms reported here, it was not used exclusively to define HER2-positivity in the assignment of receptor profiles; all patients randomized to the three arms in this report overexpressed HER2 by either IHC expression of 3+ or FISH ratio > 2.2 per the ASCO/CAP guidelines at the time the arms were running in the trial.

**Treatment.** Treatment on the T-DM1/P arm consisted of 4 cycles of T-DM1 i.v. at a flat dose of 3.6 mg/kg, given every 3 weeks, concurrent with a 840 mg i.v. loading dose of pertuzumab in week 1, followed by 420 mg every 3 weeks for 3 additional cycles. Treatment on the THP arm consisted of 12 weekly doses of paclitaxel at

80 mg/m$^2$ i.v. with concurrent trastuzumab i.v. (4 mg/kg load in week 1, followed by 2 mg/kg weekly x 11 weeks) and concurrent pertuzumab given as an 840 mg i.v. loading dose in week 1, followed by 420 mg every 3 weeks for 3 additional cycles. The TH control arm treatment consisted of 12 weekly doses of paclitaxel at 80 mg/m$^2$ i.v. and concurrent trastuzumab (4 mg/kg i.v. load in week 1, followed by 2 mg/kg weekly x 11 weeks). At the completion of this initial 12 weeks of therapy, patients on T-DM1/P, THP, and TH control arms received 4 cycles of doxorubicin (60 mg/m$^2$) and cyclophosphamide (600 mg/m$^2$) intravenously (AC), every 2–3 weeks (interval at treating physicians' discretion).

Although TH was the standard of care for HER2+ disease at the time the TDM1/P and THP arms opened, pertuzumab was granted accelerated approval in the neoadjuvant setting during the enrollment period. Randomization to the TH arm was halted, and at the direction of the DSMB, pertuzumab was added as part of standard treatment for HER2+ disease as of August 15, 2015.

Definitive surgical resection with lumpectomy or mastectomy, including management of the axilla was performed according to National Comprehensive Cancer Network guidelines[23]. Post-operative treatment was not mandated on the trial, but investigators were encouraged to give standard of care adjuvant treatment consisting of 9 additional months of trastuzumab (with or without pertuzumab, once approved), radiation (per NCCN guidelines) and endocrine therapy for those with hormone-receptor positive tumors.

**Assessments.** Core tumor biopsies, blood draws, and bilateral breast MRI were performed at baseline and at specified intervals during treatment (Supplementary Fig. 4). The primary endpoint was assessed using resected tumors by local pathologists trained in the residual cancer burden (RCB) method (in which pCR is RCB 0)[24].

As I-SPY2 is modified intent-to-treat, patients receiving any dose of study therapy are considered evaluable; those who switch to non-protocol therapy, progress, forgo surgery, or withdraw are deemed non-pCR. Secondary endpoints were RCB, and event-free and distant relapse-free survival (EFS and DRFS).

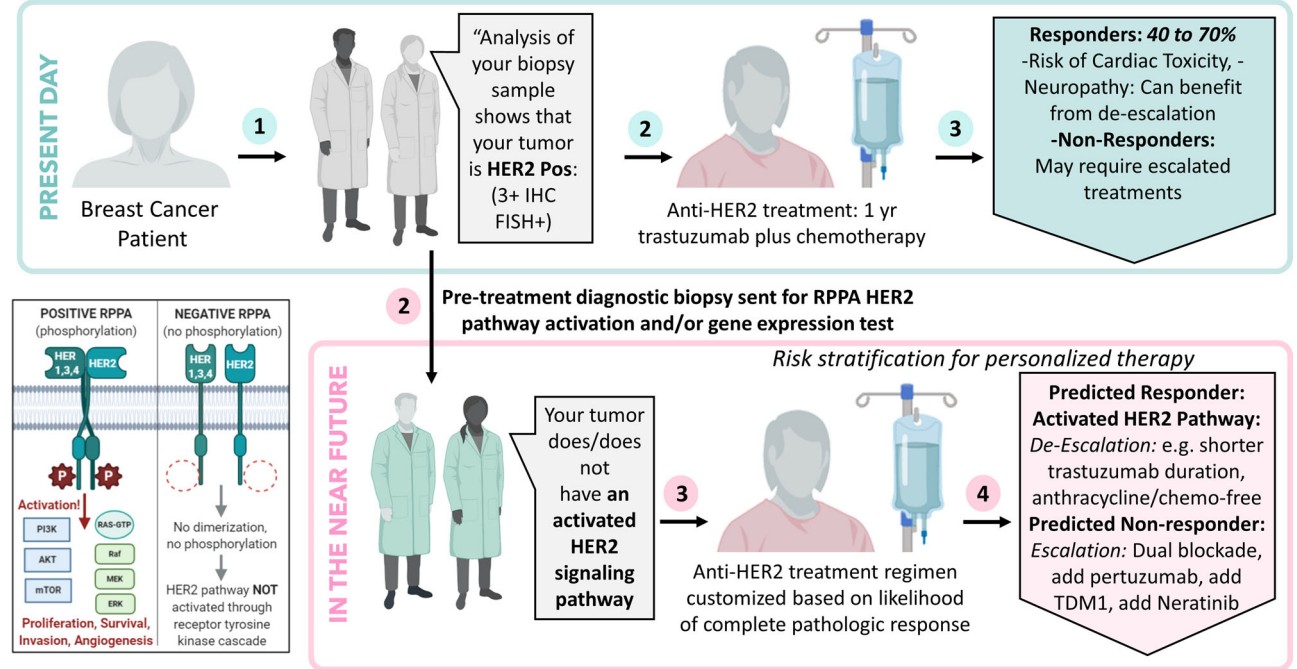

**Fig. 5 A view to the future.** Top graphic represents the current treatment paradigm: patients with HER2$^+$ breast cancer as determined by IHC or FISH receive the same neoadjuvant chemotherapy and HER2-directed therapy. Bottom graphic illustrates how implementation of the RPPA assay and gene expression profiling following positive IHC or FISH will enable de-escalation or escalation of chemotherapy and HER2-directed therapy. Original illustration: A. Haymond.

Qualified, pre-specified biomarkers evaluated included MammaPrint[17] and TargetPrint HER2 gene expression arrays using the 44 K full genome microarray (Agendia)[25], HER2 quantitative protein expression, HER2 phosphorylation (Y1248), and EGFR phosphorylation (Y1173) by RPPA. In addition, exploratory biomarker analysis used three expression signatures, PAM50 subtype, RPPA-based quantitative Ki67 expression, and Aurora kinase protein activation as biomarkers of TDM1/P or THP response.

**Trial oversight.** The I-SPY2 trial sponsor is QuantumLeap Healthcare Collaborative. Drug manufacturers supplied investigational agents and funding but played no role in study design, data accrual, data analysis, or manuscript preparation. The study complies with all local and national regulations regarding the use of human study participants and was conducted in accordance to the criteria set by the Declaration of Helsinki. The study received institutional review board approval at all clinical sites: University of California, San Diego Human Research Protections Program Institutional Review Boards, MedStar Health Research Institute-Georgetown University Oncology Institutional Review Board, Loyola University Chicago Health Sciences Division Institutional Review Board for the Protection of Human Subjects, UCSF Human Research Protection Program Institutional Review Board, UT Southwestern IRB, Chesapeake IRB, Oregon Health & Science University Research Integrity Office IRB, Mayo Clinic Institutional Review Boards, University of Pennsylvania Office of Regulatory Affairs Institutional Review Board, The University of Alabama at Birmingham Office of the Institutional Review Board for Human Use, University of Minnesota Human Research Protection Program, Colorado Multiple Institutional Review Board, Fred Hutchinson Cancer Research Center Institutional Review Board, University of Southern California Health Sciences Institutional Review Board, University of Texas MD Anderson Cancer Center Clinical Institutional Review Board, Western Institutional Review Board, University of Arizona Institutional Review Board, and Biological Sciences Division Chicago Biomedicine Institutional Review Board. An independent data and safety monitoring board (DSMB) convened monthly.

**Statistical analyses.** Probability distributions of pCR rates are calculated using a Bayesian time and covariate-adjusted logistic model with HR, HER2, and MammaPrint statuses as covariates used to calculate the probability that the pCR rate of the investigational arm is greater than control for each signature[20–22], similarly for the predictive probabilities of success in a future trial. Experimental arms are not statistically compared to each other. During the time that both the TDM-1/P and THP arms were open, pertuzumab was approved for neoadjuvant treatment in HER2$^+$ disease preventing further enrollment to the TH control arm as described above. In response, a revised statistical plan was adopted and approved by the DSMB which adjusts for time trends to allow comparison against a control population consisting of subjects enrolled since the beginning of I-SPY2.

The initial statistical analyses in I-SPY2 compared investigational arms with concurrently randomized controls. The approach applied to the first five

investigational arms: neratinib, veliparib+carboplatin, trebananib, ganitumab, and Akt inhibitor MK2206. In September 2013 the FDA granted accelerated approval for pertuzumab+trastuzumab+docetaxel as neoadjuvant therapy for high risk HER2$^+$ breast cancer. Our investigators and DSMB required dropping the I-SPY 2 control arm for HER2$^+$ subtypes because it did not contain pertuzumab, which we did by amendment in early 2014. At the time pertuzumab+trastuzumab+paclitaxel (for the first 12 weeks of neoadjuvant therapy) was an investigational arm in the trial, but it had accrued only 6 patients with none through surgery.

We wanted to be able to use the results for the original control arm but were concerned about the possibility of a drift in the prognosis of patient population over time and within patient subtype. We built a model that we call "the time machine" that adjusts for the results over time within each arm, including result for the investigational arms as well as those for control. Having multiple arms in the trial with different time periods during which they are accruing patients enabled bridging across the different eras of trial accrual. The time machine discounts results from the past, with more discounting if they are further in the past. The mathematical basis and motivation was a statistical model for bridging eras in sports[26]. The model description follows.

The control rate for an investigational arm is adjusted to the time period when the arm was being randomized to patients. Each investigational arm is compared directly against its concurrently randomized controls. The time machine strengthens this comparison by bridging to earlier controls via a series of direct comparisons. These direct comparisons are the various comparisons of arms that have been randomized in the trial, including comparisons of investigational arms against each other as well as against controls. The strength of this borrowing depends on the time-period overlaps among the various arms, both control and investigational arms. The greater uncertainty associated with results during periods of relatively low accrual and when fewer arms are being randomized is incorporated into the final analyses of the various arms.

We explicitly incorporate terms in the model to account for potential time trends in the pCR rate; we account for molecular subtype and treatment as well. This is accomplished using time-dependent offset terms in a logistic model. Time is set to 0 at each analysis. We partition time into bins of 90 days each. The index of the most recent bin, that for the previous 0–90 days, is 1. The index of the bin 91–180 days in the past is 2. And so on. Let $t_i$ be the index of the bin for the randomization time of patient $i$.

We model time-trend parameters $\delta(t)$ within each bin $t$. These are additive parameters in the model for the log-odds ratio of pCR rate for each investigational arm compared with control. We use two sets of time-trend parameters, $\delta_+(t)$ for HER2$^+$ and $\delta_-(t)$ for HER2$^-$. Consider patient $i$ who has subtype (HR$^-$, HER2$^+$, MP$^-$) and was randomized 750 days before present. Her bin $t_i$ is 9 and her time-trend offset is $\delta_+(9)$.

Suppressing subscripts + and – for both HER2$^+$ and HER2$^-$, we set $\delta(t) = 0$ for $t = 1, 2, 3, 4$. That means the previous year's results count fully in the analysis. Further in the past, that is, for $t > 4$, $\{\delta(t)\}$ is a second-order Normal Dynamic

Linear Model (NDLM)[27]. The NDLM uses the data within bins to estimate the respective log-odds ratios, but it also serves to smooth the effect across bins.

The time machine has the following structure for both HER2[+] and HER2[−], again suppressing the + and − subscripts:

$$\delta(1) = \delta(2) = \ldots = \delta(4) = 0$$

$$\delta(5) \sim N(\mu_0, \tau_0^2)$$

$$\delta(6) - \delta(5) \sim N(\mu_1, \tau_1^2)$$

$$\delta(t) - 2\delta(t-1) + \delta(t-2) \sim N(0, \tau^2) \text{ for } t > 6$$

$$\tau^2 \sim IG(\alpha, \beta)$$

In this notation, $N(\mu, \sigma^2)$ refers to a normal distribution with mean $\mu$ and standard deviation $\sigma$ and $IG$ stands for inverse gamma. The parameters of the prior distributions are $\mu_0 = \mu_1 = 0$, $\tau_0^2 = \tau_1^2 = 0.001$, $\alpha = 1$, and $\beta = 0.001$.

An exploratory analysis of EFS was performed using patients with follow-up data as of February 26, 2019. EFS was assessed as time from treatment consent to any locoregional or distant recurrence or death from any cause; and patients without events were censored at last follow-up. Kaplan–Meier survival curves of each arm were prepared; Cox proportional hazard modeling was used to estimate hazard ratios between T-DM1/P or THP and control arms for each signature. Statistics from this exploratory analysis are descriptive and not inferential–sample sizes are small within signatures and I-SPY2 is not powered for EFS or other survival endpoints.

**Biomarker analysis.** Biomarker analyses were performed on pre-treatment biopsies, which included the Agendia 44 K full-genome microarray and reverse-phase protein array (RPPA, performed by author EP at George Mason University). We evaluated HER2 IHC, 3 expression signatures, PAM50 subtype, and 5 protein/phospho-protein endpoints by RPPA as biomarkers of TDM1/P response.

HER2 expression by immunohistochemistry (IHC) was scored per ASCO-CAP HER2 testing guidelines (positive: 3+, equivocal: 2+, negative:1+). Pre-treatment tumor samples were assayed using Agilent 44 K (32627) or 32 K (15746) expression arrays; and these data were combined into a single gene-level dataset after batch-adjusting using ComBat[28]. ERBB2 amplicon gene expression (Mod7_ERBB2) and proliferation (Module11_Prolif) modules were scored as published[29]. The estrogen signature, ER_PGR_ave, was scored as the average of ESR1 and PGR expression. PAM50 'intrinsic' subtype classifications were evaluated using the published method and R scripts by J. Parker[30], applied to expression data from 1151 patients screened for I-SPY2 (on study: 986; low risk registry: 165), with class assignments with confidence level < 0.8 treated as NA's. These data were centered to a 1:1 ratio of HR[+] to HR[−] samples prior to classification.

In addition, LCM was performed to isolate tumor epithelium for signaling protein activation profiling by reverse phase protein arrays (RPPA). Protein/phospho-protein endpoints reflecting HER-family signaling (ERRB2 total protein, ERBB2 Y1248 and EGFR Y1173) and proliferation (Ki67 total protein and Aurora A T288/B T232/C T198) were selected for further analysis. All TDM1 samples were on the same RPPA array, but the controls were distributed over 3 RRPA arrays. To put the data on the same scale, we z-score normalized the HER2[+] patient subset of each array and batch-adjusted using ComBat before combining into a single dataset. The consort diagram with the number of evaluable patients for each molecular profiling analysis is shown in Figure Y. Details of the sample preparation and data processing are as previously described[13].

Pre-specified analyses used logistic modeling to assess biomarker performance within each arm with significance assessment using the likelihood ratio (LR) test (one-tailed test of LR statistic against a chi-square distribution with 1 degree of freedom, $p < 0.05$). These analyses were also performed adjusting for HR status as a covariate, and within receptor subsets, sample size permitting. Associations between categorical variables (e.g., HER2 IHC level and PAM50 subtype) were assessed using Fisher's exact test. Biomarkers were assessed individually without adjustment for multiple hypothesis testing. All computation was performed in the R programming environment (version 3.3.3).

**Reporting summary.** Further information on research design is available in the Nature Research Reporting Summary linked to this article.

## Data availability
Source data are provided with this paper for all outcomes except adverse events in Supplementary Data 1. A summary of all adverse events observed are provided in Supplementary Data 2. In addition, gene expression and base clinical variables have been deposited in GEO under accession number GSE181574.

## Code availability
Biomarker data analysis was performed using R version 3.6.3 and Bioconductor ver.3.10; code available upon request from ispyadmin@ucsf.edu. The randomization engine and Bayesian analytic software used in efficacy analysis are used under license from Berry Consultants, LLC; requests for code should be directed to don@berryconsultants.com.

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

## Acknowledgements

Supported by Quantum Leap Healthcare Collaborative (2013 to present) and the Foundation for the National Institutes of Health (2010 to 2012), a grant from the Gateway for Cancer Research (G-16-900), and by a grant (28XS197) from the National Cancer Institute Center for Biomedical Informatics and Information Technology. The authors sincerely appreciate the ongoing support for the I-SPY2 TRIAL from the Safeway Foundation, the William K. Bowes, Jr. Foundation and Give Breast Cancer the Boot. Initial support was provided by Quintiles Transnational Corporation, Johnson & Johnson, Genentech, Amgen, the San Francisco Foundation, Eli Lilly, Pfizer, Eisai Co., Ltd., Side Out Foundation, Harlan Family, the Avon Foundation for Women, Alexandria Real Estate Equities, and private individuals and family foundations. The study sponsor, Quantum Leap Healthcare Collaborative, provided administrative, operational and software support for the design, data collection, and analysis of this study.

We thank Anna Barker for leadership in helping to launch the I-SPY2 trial, the members of the data and safety monitoring committee, the trial coordinators, Ken Buetow and the staff of caBIG for input with the informatics design, the entire project oversight committee and the many investigators who have contributed. We are grateful for the input of our wonderful patient advocates: Susie Brain, Thelma Brown, Elly Cohen, Deborah Collyar, Coleen Crespo, Amy Delson, Peggy Devine, Sandra Finestone, Elizabeth Frank, Diane Heditsian, Patricia Haugen, Deborah Laxague, Marisa Leonardelli, Barbara LeStage, Beverly Parker, Susan Samson and Patty Spears. Thank you to all the patients who volunteered to participate in I-SPY2.

## Author contributions

Conceptualization: L.E., D.B., L.V., D.Y., J.P., W.F.S., N.H., D.B., A.D.; Methodology: L.E., D.B., C.Y., D.W., E.P., L.V., L.B.S., R.G., S.B., G.H., N.H., W.F.S.; Formal analysis: C.Y., D.W., E.P., A.S., J.W., D.B.; Investigation: A.C., C.Y., D.W., E.P., S.M., A.W., A.J.C., C.I., K.A., D.T., E.E., A.F., R.Y., H.H., J.L., R.V., B.H., K.K., A.E., R.N., J.B., J.W., L.B., R.G., T.H., E.R., C.E., M.A., E.C., P.R., M.B., S.A., M.M., H.R., R.S., W.F.S., N.H., D.Y., L.V., L.E., A.D.; Data curation: C.Y., E.P., L.V., J.W., L.B.S., R.G., M.B., J.C., M.P., S.A., A.W., G.H., R.S., K.S., A.A., J.M., A.S.; Original draft: A.C., C.Y., D.W., E.P., A.D.; Writing (review and editing): A.C., C.Y., D.W., E.P., J.M., H.R., L.E., A.D.; Visualization: A.C., C.Y., D.W., E.P., A.W., J.M., A.S., A.D.; Supervision: L.V., D.Y., J.P., H.R., R.S., W.F.S., N.H., D.B., L.E., A.D.; Project administration: M.B., J.C., M.P., S.A., G.H., R.S., K.S., A.A., L.E., A.D.; Funding acquisition: L.E.

## Competing interests

Amy S. Clark has received unrelated research support from Novartis. Christina Yau has consulted for NantOmics, LLC, part of the NantWorks network. E.F. Petricoin reports leadership roles with: Perthera and Ceres Nanosciences; stock or other ownership interests in Perthera, Ceres Nanosciences and Avant Diagnostics; consulting or advisory roles with Perthera, Ceres Nanosciences, AZGen, Avant Diagnostics; institutional research support from Ceres Nanosciences, GlaxoSmithKline), Abbvie, Symphogen, and Genentech; patents, royalties, other intellectual property from National Institutes of Health patents licensing fee distribution/royalty, co-inventor on filed George Mason University–assigned patents related to phosphorylated HER2 and EGFR response predictors for HER family-directed therapeutics, as such can receive royalties and licensing distribution on any licensed IP; travel, accommodations, expenses received from Perthera and Ceres Nanosciences. Laura J. van 't Veer is a part-time employee and stockholder in Agendia N.V. Doug Yee has received unrelated research support from Boehringer Ingleheim. StacyMoulder has been an employee of Eli Lilly since 2021. A. Jo Chein reports institutional research support from Seagen, Merck, Amgen and Puma Biotechnology. Claudine Isaacs has received consulting fees from Seattle Genetics, Genentech, AstraZeneca, Novartis, PUMA, Pfizer, and Esai. Judy C. Boughey has received support from Eli Lilly. Barbara Haley reports that UTSouthwestern Medical Center Dallas receives funding for her research as PI from Pfizer, Lilly, Daiichi Sankyo, Roche, Puma, Astra Zeneca and Sanofi. Hyo S. Han reports institutional research support from GlaxoSmithKline, Abbvie, Prescient, G1 therapeutics, Marker therapeutics, Novartis, Horizon Pharma, Pfizer, Seattle Genetics, Arvinas and Zymeworks; she has received a grant from the Department of Defence and is a member of Lilly' Speaker's Bureau. Andres Forero-Torres became a Seattle Genetics employee in 2018 and holds stock option from this employment. Julie Lang has previously received a research grant from ANGLE Parsortix and has received honoraria as part of the Genomic Health's speakers' bureau and a Puma Biotechnology advisory board. Debu Tripathy receives research support from Novartis, Pfizer, Polyphor and has been a paid consultant for services on steering committees or advisory boards for Novartis, Pfizer, AstraZeneca, GlaxoSmithKline, OncoPep, Gilead, Exact Sciences. Rita Nanda has received research support from Arvinas, AstraZeneca, Celgene, Concept Therapeutics, Genentech/Roche, Immunomedics/Gilead, Merck, OBI Pharm, Inc., Odonate Therapeutics, OncoSec, Pfizer, Taiho and SeaGen. J.D. Wulfkuhle received honoraria from DAVA Oncology and consults for Baylor College of Medicine. Melissa Paolini is an employee and stock holder in Arcus Biosciences. Donald Berry and Scott Berry are co-owners of Berry Consultants, LLC, a company that designs adaptive Bayesian clinical trials (including I-SPY 2) for pharmaceutical and medical device companies, NIH cooperative groups, patient advocacy groups, and international consortia. Hope Rugo reports institutional research support from Pfizer, Merck, Novartis, Lilly, Genentech, Odonate, Daiichi, Seattle Genetics, Eisai, Macrogenics, Sermonix, Boehringer Ingelheim, Polyphor, Astra Zeneca and Immunomedics, and has received honoraria from Puma Biotechnology, Mylan and Samsung. W. Fraser Symmans is a co-founder with equity in Delphi Diagnostics that licensed intellectual property, is a co-inventor of an issued patent for the algorithm to calculate residual cancer burden that is freely available on the internet, holds publicly traded shares in IONIS Pharmaceuticals and Eiger Biopharmaceuticals, and is an unpaid scientific advisor to Roche for translational research related to the KAITLIN trial and unpaid steering committee member for the KATHERINE trial. Laura Esserman is an unpaid member of the board of directors of Quantum Leap Healthcare Collaborative (QLHC) and received grant support from QLHC for the I-SPY2 Trial; she is a member of the Blue Cross/Blue Shield Medical Advisory Panel and receives reimbursement for her time and travel; Dr. Esserman has received unrelated research support from Merck. Angela DeMichele has received honoraria or consulting fees from Pfizer and Context Therapeutics and reports institutional research support from Novartis, Pfizer, Genentech, Calithera and Menarini. All other authors declare no competing interests.
