## [Peer Review File · Nature Communications]

Reviewers' Comments:

Reviewer #2:

Remarks to the Author:

This is a well written report focusing on biomarkers and response to HER2 -directed therapy in the new adjuvant setting using the I-SPY platform. The biomarker discussion was the most interesting part of the paper and of the data presented. Actually the presentation of the clinical information could be condensed somewhat....most of this data including toxicity is well known from other trials and clinical experience. Similarly as numbers are small in each arm, the issue of long term outcome speculation can also be abbreviated as it is largely speculative. The numbers of CNS events are so small that a paragraph on that in the discussion could also be deleted.

It may be worth adding a sentence or two for the uninitiated....regarding the Kristine trial. How the results in this experience square with Kristine trial

Although the discussion does speculate on how the biomarker information could be used....might be helpful to create a figure that speculatesaspirationally...how such information could be used in the future for clinical decision making

Reviewer #3:

Remarks to the Author:

This manuscript compares TDM1/P and THP to TH in the I-SPY2 trial demonstrating superiority of each of the experimental therapies to control. Statistically, there is not much to comment on since the methods for evaluation in I-SPY2 have been very well developed. The evaluations use underlying Bayesian models giving plausible distributions for pCR rates adjusting in the background for significant covariates (hormone receptor status; PAM50 classification) when needed. These are then used to predict success in Phase III trials (graduation defined as 85% probability or better). The comparison to the control arm does include a wider randomization period for the control arm which is then adjusted by time trends (cleverly called The Time Machine in the Supplement). Thus, one may see more discrepancies in baseline characteristics across the arms.

The Supplement is extremely important as it contains critical information on comparisons of the biomarkers by pCR outcome. Table 3 is most important here – personally would have preferred that table in the manuscript instead of the EFS curves with limited information. The text often refers to differences in these markers without much sense of the absolute difference and the associated differences in likelihood ratio p-values.

Overall there are some impressive differences in pCR rates and biomarkers by pCR given the limited sample sizes. The manuscript is well written and provides clinically useful data.

Below are some minor points to consider:

1. I-SPY2 does not compare experimental arms, but now that THP has become the new control arm (dropping TH) does that then allow a comparison of TDM1/P to THP? This would be a contemporaneous comparison as well.
2. EFS Kaplan-Meier curves are considered descriptive, but the hazard ratio is given without giving a confidence/credibility interval. The CI would be extremely wide showing that the hazard ratio is not a reliable estimate and should be omitted with so few events. Similarly, the reported EFS survival rates are not all that reliable either. Figure 3 could be moved to the Supplement.
3. It was unclear if the starting point for EFS was randomization since of course pCR cannot be determined until later while no pCR can be immediately apparent at relapse. If these were landmarked analyses it would avoid the possibility of immortal time bias. This was not clear in the statistical section.
4. Results. Efficacy. The overall pCR rates are driven by the mix of HR+ to HR-. Those were described in the text while the HR specific values were not. The latter seem more useful. Also the first time "three HER2-positive signatures" was used it may confuse the reader since the Methods follow at the end. I was expecting three mutually exclusive signatures.

Reviewer #1 (Remarks to the Author):

This is a well written report focusing on biomarkers and response to HER2 -directed therapy in the new adjuvant setting using the I-SPY platform. The biomarker discussion was the most interesting part of the paper and of the data presented.

- 1) Actually the presentation of the clinical information could be condensed somewhat....most of this data including toxicity is well known from other trials and clinical experience.
Response: We thank this reviewer for the suggestion and have reduced our discussion and presentation of the clinical information.
- 2) Similarly as numbers are small in each arm, the issue of long term outcome speculation can also be abbreviated as it is largely speculative.
Response: We thank the reviewer for this suggestion. We have shortened the paragraph discussing EFS.
- 3) The numbers of CNS events are so small that a paragraph on that in the discussion could also be deleted.
Response: We appreciate this suggestion. While we agree that we can the number of CNS events are small, CNS recurrence is a huge challenge in HER2 positive breast cancer. We did shorten our discussion about the CNS events (page 11).
- 4) It may be worth adding a sentence or two for the uninitiated....regarding the Kristine trial. How the results in this experience square with Kristine trial
Response: We agree that discussion of Kristine will add to our manuscript. We added two sentences in the discussion at the end of the first paragraph (end page 7 and top page 8).
- 5) Although the discussion does speculate on how the biomarker information could be used....might be helpful to create a figure that speculatesaspirationally...how such information could be used in the future for clinical decision making
Response: We thank the reviewer for this idea. We have added a figure (Figure 5) which demonstrates our aspirational goal of how we envision using this information in clinical decision making.

Reviewer #2 (Remarks to the Author):

This manuscript compares TDM1/P and THP to TH in the I-SPY2 trial demonstrating superiority of each of the experimental therapies to control. Statistically, there is not much to comment on since the methods for evaluation in I-SPY2 have been very well developed. The evaluations use underlying Bayesian models giving plausible distributions for pCR rates adjusting in the background for significant covariates (hormone receptor status; PAM50 classification) when needed. These are then used to predict success in Phase III trials (graduation defined as 85% probability or better). The comparison to the control arm does include a wider randomization period for the control arm which is then adjusted by time trends (cleverly called The Time Machine in the Supplement). Thus, one may see more discrepancies in baseline characteristics across the arms.

The Supplement is extremely important as it contains critical information on comparisons of the biomarkers by pCR outcome.

- 1) Table 3 is most important here – personally would have preferred that table in the manuscript instead of the EFS curves with limited information. The text often refers to differences in these markers without much sense of the absolute difference and the associated differences in likelihood ratio p-values.

Response: We appreciate this point. We have removed the original Figure 3 from the manuscript and have moved the original Supplementary Table 3 into the main manuscript (now Table 4 in the revised manuscript).

Overall there are some impressive differences in pCR rates and biomarkers by pCR given the limited sample sizes. The manuscript is well written and provides clinically useful data.

Below are some minor points to consider:

- 2) I-SPY2 does not compare experimental arms, but now that THP has become the new control arm (dropping TH) does that then allow a comparison of TDM1/P to THP? This would be a contemporaneous comparison as well.

Response: Although THP has become the new (bridging) control in I-SPY 2, it was an experimental regimen during the same time period as TDM1/P; and we cannot conduct a contemporaneous comparison between these arms. It would be possible to provide a comparison between THP and TDM1/P based on non-contemporaneous bridging THP control patients (using the time machine); but we feel that this comparison is less relevant and may distract from the main focus of the manuscript. We note that the posterior distribution of pCR probabilities summarized in Figure 2 came from the same Bayesian covariate-adjusted logistic model with time trend adjustments. While we cannot directly compare the two experimental arms, the figure is aligned such that readers can see the pCR probability distributions THP and TDM1/P in the context of each other (but in comparison to the TH control).

- 3) EFS Kaplan-Meier curves are considered descriptive, but the hazard ratio is given without giving a confidence/credibility interval. The CI would be extremely wide showing that the hazard ratio is not a reliable estimate and should be omitted with so few events. Similarly, the reported EFS survival rates are not all that reliable either. Figure 3 could be moved to the Supplement.

Response: As noted above, we have removed Figure 3 from the manuscript altogether, since the data is presented in a more detailed fashion in Supplementary Figure 1.

- 4) It was unclear if the starting point for EFS was randomization since of course pCR cannot be determined until later while no pCR can be immediately apparent at relapse. If these were landmarked analyses it would avoid the possibility of immortal time bias. This was not clear in the statistical section.

Response: We thank the reviewer for pointing this out. Consistent with our previous publication of the I-SPY 2 EFS data (Yee et al, 2020), the start time of EFS was calculated from the time of treatment consent. The following sentence have been added to the methods section to clarify this: “EFS was assessed as time from treatment consent to any locoregional or distant recurrence or death from any cause; and patients without events were censored at last follow-up.”

We did not perform landmarked analyses to address the possibility of immortal time bias. At the reviewer’s suggestion, we evaluated using the time to surgery (for each individual patient) as the landmark time and observed that two patients (one pCR patient

on the TDM1/P arm and 1 non-pCR patient from the THP arm) would have been removed if we were to perform a landmarked analysis. However, since our EFS analyses were exploratory, for consistency with our previous publication, we did not change the results presented in the manuscript (Supplemental Figure 1). Follow up for all 3 arms is still ongoing, and may elucidate further in the future as more EFS events occur, particularly in the HR-positive/HER2-positive group.

- 5) Results. Efficacy. The overall pCR rates are driven by the mix of HR+ to HR-. Those were described in the text while the HR specific values were not. The latter seem more useful. Also the first time “three HER2-positive signatures” was used it may confuse the reader since the Methods follow at the end. I was expecting three mutually exclusive signatures.

Response: We thank the reviewer for pointing out this area of confusion. Clarification and identification of these three signatures is now provided in the last paragraph of the introduction (top of page 4).

Reviewers' Comments:

Reviewer #3:

Remarks to the Author:

The authors have adequately addressed the prior issues. No further comments.